# Establishing Normative Values for Acromion Anatomy: A Comprehensive MRI-Based Study in a Healthy Population of 996 Participants

**DOI:** 10.3390/diagnostics14010107

**Published:** 2024-01-03

**Authors:** Anne Prietzel, Theo Languth, Robin Bülow, Till Ittermann, René Laqua, Lyubomir Haralambiev, Georgi Iwan Wassilew, Axel Ekkernkamp, Mustafa Sinan Bakir

**Affiliations:** 1Center for Orthopaedics, Trauma Surgery and Rehabilitation Medicine, University Medicine Greifswald, Ferdinand-Sauerbruch-Straße, 17475 Greifswald, Germany; anne.prietzel@med.uni-greifswald.de (A.P.);; 2Institute of Diagnostic Radiology and Neuroradiology, University Medicine Greifswald, Ferdinand-Sauerbruch-Straße, 17475 Greifswald, Germany; 3Institute for Community Medicine, University Medicine Greifswald, Ferdinand-Sauerbruch-Straße, 17475 Greifswald, Germany; 4Institute of Diagnostic Radiology, Städtisches Krankenhaus Kiel, Chemnitzstraße 33, 24116 Kiel, Germany; 5Department of Trauma Surgery and Orthopedics, BG Hospital Unfallkrankenhaus Berlin gGmbH, Warener Straße 7, 12683 Berlin, Germany

**Keywords:** shoulder pain, acromion anatomy, acromioplasty, subacromial impingement, rotator cuff lesions, MRI diagnostics, reference values

## Abstract

Shoulder pain is a common issue often linked to conditions such as subacromial impingement or rotator cuff lesions. The role of the acromion in these symptoms remains a subject of debate. This study aims to establish standardized values for commonly used acromion dimensions based on whole-body MRI scans of a large and healthy population and to investigate potential correlations between acromion shape and influencing factors such as sex, age, BMI, dominant hand, and shoulder pain. The study used whole-body MRI scans from the Study of Health in Pomerania, a German population-based study. Acromion index, acromion tilt, and acromion slope were measured. Interrater variability was tested with two independent, trained viewers on 100 MRI sequences before actual measurements started. Descriptive statistics and logistic regression were used to evaluate the results. We could define reference values based on a shoulder-healthy population for each acromion parameter within the 2.5 to 97.5 percentile. No significant differences were found in acromion slope, tilt, and index between male and female participants. No significant correlations were observed between acromion morphology and anthropometric parameters such as height, weight, or BMI. No significant differences were observed in acromion parameters between dominant and non-dominant hands or stated pain intensity. This study provides valuable reference values for acromion-related parameters, offering insight into the anatomy of a healthy shoulder. The findings indicate no significant differences in acromion morphology based on sex, weight, BMI, or dominant hand. Further research is necessary to ascertain the clinical implications of these reference values. The establishment of standardized reference values opens new possibilities for enhancing clinical decision making regarding surgical interventions, such as acromioplasty.

## 1. Introduction

Shoulder pain is a prevalent issue affecting many patients [1], often attributed to conditions such as subacromial impingement or rotator cuff lesions [2,3]. The role of the acromion in the development of these symptoms in atraumatic cases remains a subject of conflicting debate [4,5,6,7]. Moreover, the lack of a standardized method to describe acromion shape further complicates this discussion.

One of the earliest and most commonly used classifications was proposed by Bigliani et al. in 1986 [8]. They identified three types of acromial morphology in outlet-view radiographs: flat (type I), curved (type II), and hooked (type III) acromion. The hooked type was later associated with a higher likelihood of developing shoulder symptoms [9,10]. However, contrasting findings from other authors have questioned the correlation between the acromion type according to Bigliani and shoulder pathologies [9]. Some did not detect any correlation between the acromion type according to Bigliani and shoulder pathologies [11]. Moreover, the types appear to be independent of age or sex [9,12,13].

A major limitation of the Bigliani classification is its poor interrater reliability, leading to the development of alternative measurements to describe acromion morphology [14]. These include the acromial slope [8,15], acromial tilt [15], acromion index [16], and the lateral acromion angle [17].

Despite their improved reliability, these measurements lack well-defined reference values for each parameter, making it challenging to establish a standard acromion appearance. Reference values are essential for clinical assessment to determine if a patient’s symptoms may be attributed to acromion shape, influencing decisions regarding surgical interventions like acromioplasty [7,18,19].

While the mentioned acromion measurements are based on X-ray assessments, magnetic resonance imaging (MRI) is considered the gold standard in shoulder pain diagnostics [20]. Thus, the objective of this study is to establish standardized values for commonly used acromion dimensions based on whole-body MRI scans of a large and healthy population. Additionally, we aim to investigate potential correlations between acromion shape and influencing factors such as sex, age, BMI, dominant hand, and shoulder pain. By addressing these research objectives, we aim to contribute valuable insights to the understanding of acromion anatomy and its implications for shoulder pathologies.

## 2. Materials and Methods

### 2.1. SHIP Design

The study of Health in Pomerania is a German population-based study performed in the counties of Northern and Eastern Pomerania and the German cities Stralsund and Greifswald to describe the health status of the Pomeranian society [21,22]. There were two independent cohorts of Pomeranian citizens examined aged from 20 to 79 years. In 1997, 6265 citizens were randomly chosen from the official resident registry office and invited via three letters, phone calls, and one personal contact. Migrated and deceased persons were excluded. The first cohort comprised 4308 participants (response 68.8%) from 1997 to 2001 who underwent an interview as well as several non-invasive examinations. The standardized interview included pain-related questions, e.g., if the participants suffer from neck or shoulder pain.

Two follow-up examinations were executed from 2002–2006 (SHIP-1, *n* = 3300) and 2008–2012 (SHIP-2, *n* = 2333). In 2008, there was another independent cohort accomplished (SHIP-TREND, *n* = 4420, response 50%) with a wider spectrum of examinations [21,22]. Overall, 3371 of 6753 experimentees from SHIP-2 and SHIP-TREND went through a whole-body MRI examination (Magnetom Avanto, 1.5 Tesla, Siemens Healthcare, Erlangen, Germany), which were used for the current study. The whole-body MRI was performed as a TIRM fat-suppressed imaging with a voxel size of 2.1 × 1.6 × 5 mm. The patients were lying in the MRI in a supine position; head first, on the table, the arms were positioned laterally lying at their sides. Reasons for dropout were, for example, claustrophobia, metal implants, or personal reasons. MRI scans were evaluated by two independent radiologists with the help of a standardized examination sheet. SHIP was the first population-based study using whole-body MRI examinations. The complete MR procedure and imaging protocol including further technical data have already been published [23].

### 2.2. Image Analysis and Measurement

In total, there were 3371 whole-body MRI sequences to be measured. Horos (Horos Project Community, GNU Lesser General Public License, v. 3.3.5) was used as an image viewer in addition to a plug-in which was prepared especially for these measurements. It converted single points in the MRI sequences into three-dimensional distances and angles. In detail, these are the three parameters: acromion index, acromion tilt, and acromion slope. A single observer analyzed the MRI sequences blinded to any subjects’ health data.

### 2.3. Acromion Index (Figure 1)

The index is the relation of the distances between the articular surface of the glenoid to the lateral edge of the humeral head, on the one hand, and the lateral edge of the acromion on the other hand. Measuring took place by an adjusted form of the method by Nyffeler et. al. [16], but in a horizontal plane.

**Figure 1 diagnostics-14-00107-f001:**
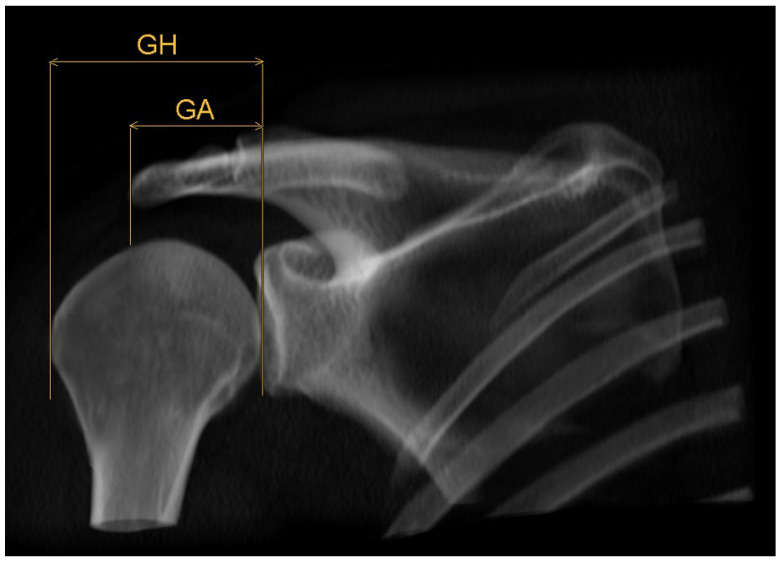
Schematic 3D reconstruction of the glenohumeral joint in anterior–posterior view, illustrating the measurement technique for the acromion index AI = GA/GH. GA indicates the distance between glenoid surface and lateral edge of the acromion, GH the distance between glenoid surface and lateral edge of the humeral head.

First, pinpoints are set to mark the lower (G1, Figure 2) and the upper (G2, Figure 3) end of the glenoid, each in the middle of it. A line connecting these two points, created by the plug-in, representing the axis of the glenoid. Another point H (Figure 4) marks the lateral edge of the humeral head.

The plug-in calculates the distance GH between the glenoid surface at the same height as point H is marked. Afterwards, another point A (Figure 5) shows the lateral edge of the acromion, from which distance GA is calculated by the plug-in in the same way. The acromion index is the relation between both distances, so AI = GA/GH.

### 2.4. Acromion Tilt (Figure 6)

The acromion tilt shows the tilt angle of the acromion in relation to the coracoid process in the sagittal plane. Measuring was based on the method by Kitay et. al. [15], but in the horizontal plane. Two points mark the ventral (A1, Figure 7) and the dorsal (A2, Figure 8) end of the acromion’s bottom. A line connecting these points builds line A. 

**Figure 6 diagnostics-14-00107-f006:**
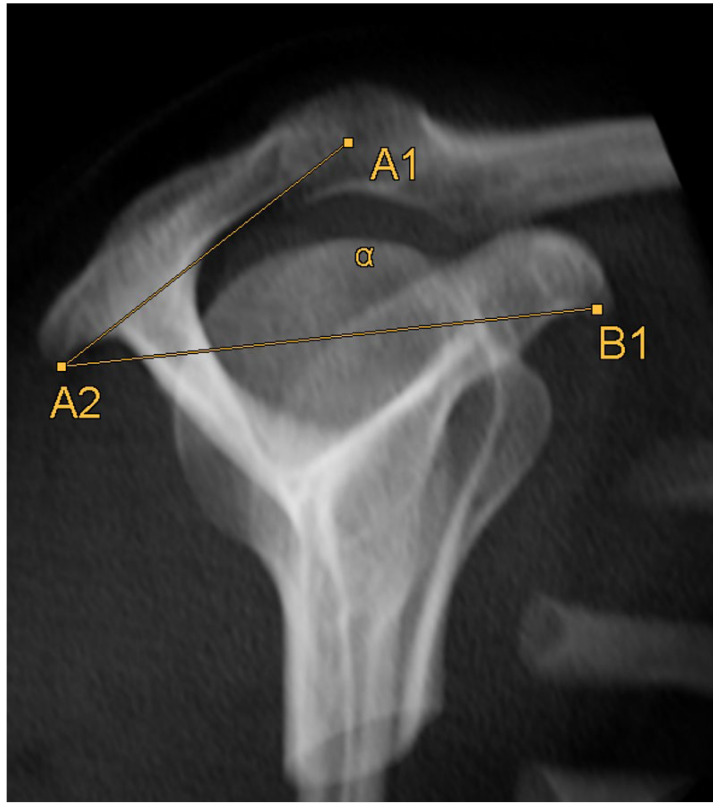
Schematic 3D reconstruction of the glenohumeral joint in anterior–posterior view, illustrating the measurement technique for the acromion tilt (α). A1 indicates the ventral end of the acromion’s bottom, A2 the dorsal end of the acromion’s bottom, B1 the lower end of the coracoid process.

Afterwards, a point B1 (Figure 9) is set on the lower end of the coracoid process. A line connecting point B1 with point A2 builds line B. The plug-in calculates the angle between lines A and B, which shows the acromial tilt.

### 2.5. Acromion Slope (Figure 10)

The acromion slope measures the curvature of the acromion in the sagittal plane. Measuring was based on the method by Bigliani et al. [8] and Kitay et al. [15], but once more in the horizontal plane. One line connects the ventral end of the acromion (A1, Figure 7) with the middle of the acromion’s bottom (M, Figure 11). Another line connects the dorsal end (A2, Figure 8) with the middle of the acromion’s bottom (M). The plug-in takes points A1 and A2 from the acromion tilt and calculates the angle between those two lines.

**Figure 10 diagnostics-14-00107-f010:**
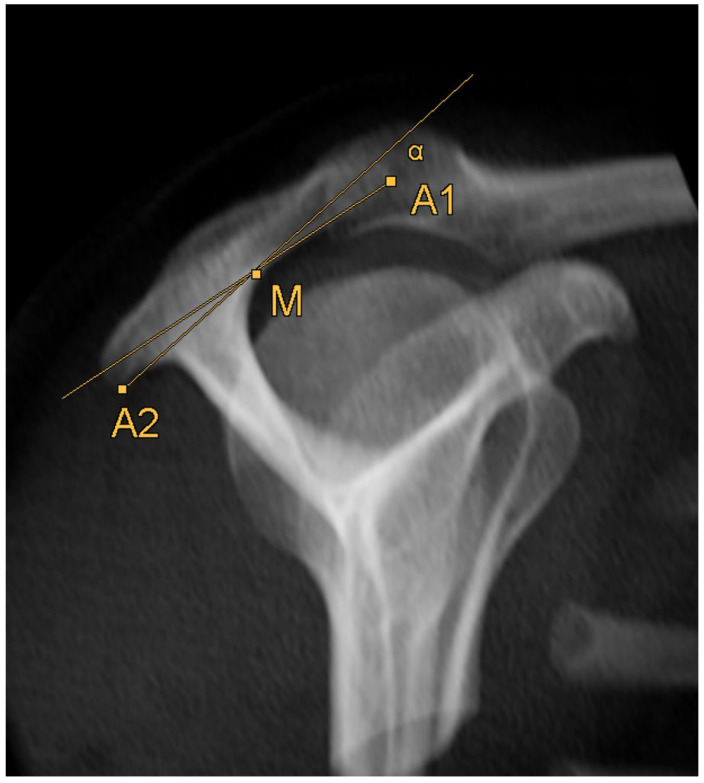
Schematic 3D reconstruction of the glenohumeral joint in anterior–posterior view, illustrating the measurement technique for the acromion slope (α). A1 indicates the ventral end of the acromion’s bottom, A2 the dorsal end of the acromion’s bottom, M the middle of the acromion’s bottom.

### 2.6. Statistical Analysis

Interrater variability was tested on 100 MRI sequences before actual measurements started. Two independent, trained viewers used the plug-in to validate the method. Bland and Altman plots as well as intraclass correlation coefficients were used to evaluate intra- and interreader variabilities (Table 1 and Figure 12).

Descriptive statistics like mean values, standard deviations (SDs), ranges, and percentiles were used to evaluate the results. Logistic regression adjusted for age and sex was used to investigate the correlation between acromion parameters and the prevalence of pathological changes like edema, peripheral attachments, and pain. Pearson’s correlation coefficient was used to determine the correlation between the right and left side as well as possible interactions between the parameters.

Stratified by sex, upper and lower reference limits were calculated by quantile regressions for the 2.5th and the 97.5th percentile in the same way the influence of age, weight, and height was studied. We excluded patients who showed an edema, joint effusion, or bony peripheral attachments and those who declared a pain intensity bigger than 3 on a scale from 0 to 10 or a fracture in the close anatomical region of the shoulder (15.18%) for the group of reference values.

A *p*-value less than 0.05 was considered statistically significant. We assumed small impact for a Pearson’s correlation coefficient ≥ 0,1, moderate impact for beta ≥ 0.3, and big impact for beta ≥ 0.5 [24]. Statistical analysis was performed using common statistical software: Stata 17.0 (Stata Corporation, College Station, TX, USA).

## 3. Results

From the initial 3371 whole-body MRI sequences, only 1034 sequences showed clear visualization of the acromioclavicular joint (Figure 13). Thirty-eight sequences were excluded due to blurring. In the MRI scans of the remaining 996 participants, we were able to analyze the acromion 915 times on the right side and 642 times on the left side.

### 3.1. Correlation Analysis

#### 3.1.1. Comparison between Left and Right Side

A significant correlation coefficient of 0.53 was found between the acromion tilt measurements on the left and right sides, indicating at least a moderate association. The acromion index exhibited a lesser moderate correlation (coefficient of 0.34), while the acromion slope showed no significant correlation (coefficient of 0.19) between both sides (Table 2).

#### 3.1.2. Comparison between Genders

No significant differences were found in acromion slope, tilt, and index between male and female participants (Table 3).

#### 3.1.3. Correlation with Anthropometric Parameters

No significant correlations were observed between acromion morphology and anthropometric parameters such as height, weight, or BMI (Table 4).

#### 3.1.4. Hand Dominance and Pain Intensity

Concerning the dominance of one hand, 90.53% of participants were right-handed, 4.62% left-handed, and 4.85% ambidextrous. No significant differences were observed in acromion parameters between dominant and non-dominant hands (Table 5).

No significant differences were observed in acromion parameters between participants with pain intensity greater than 3 on a scale from 0 to 10 (15.18% of all participants) and those with an intensity of 3 or lower (Table 6).

#### 3.1.5. Reference Values

We defined our exclusion criteria to estimate the standard values for a healthy population. Reference values for an average acromion in a healthy shoulder were calculated within the 2.5th to 97.5th percentile for all participants, as there were no significant differences between men and women (Table 7).

## 4. Discussion

In this study, we have presented reference values for acromion-related parameters, which, to the best of our knowledge, are reported for the first time. Despite MRI not being the gold standard for examining bony structures, our results show good comparability with previous studies using CT or X-ray imaging, particularly regarding acromion tilt [19,25]. For instance, Balke et al. using a.p. and outlet-view X-ray reported acromion slope values of 21–25°, acromion tilt values of 29–34°, and an acromion index range of 0.67–0.75 [10]. Nyffeler et al. described an acromion index of 0.64 in a.p. X-rays [16], while Chaimongkhol et al. discovered an acromion tilt of 28° through manual measurements [13].

MRI proves to be a valid alternative diagnostic tool for assessing acromial issues, as demonstrated by Chalmers et al. without any radiation exposure [19]. Additionally, our measurements seem transferable to other diagnostic imaging methods. However, radiography is often more readily available and faster, making it a reasonable addition in the diagnostics of acromial pathologies, such as subacromial impingement syndrome. Differences in acromion slope and index parameters between MRI and X-ray diagnostics might be attributed to varying interobserver reliability in interpreting the imaging and differences in the imaging planes used as well as the different positioning of the scapula during radiography since MRI was conducted in a supine position, in contrast to X-ray imaging in an erect position. The advantage of MRI lies in its more consistent image quality due to the possibility of reconstruction techniques, potentially resulting in higher accuracy of reference values compared to X-ray diagnostics.

An interesting finding is the lack of correlation of the acromion parameters with one another, suggesting that it is not feasible to classify acromion shapes into distinct “types” based on acromion tilt, slope, and index. We could not reproduce the historical classification according to Bigliani et al. using a shoulder outlet-view radiograph, which may have yielded different findings [8]. It has been previously reported that the Bigliani classification has poor interobserver reliability, leading to its decreasing significance [14]. While our study did not directly investigate pathological conditions, and thus did not utilize the Bigliani classification for data analysis, we recognize that the morphological characteristics we measured may have implications for understanding shoulder pathologies. Although there are limitations of the Bigliani classification, notably its poor interrater reliability, we suggest that future research could explore the relationship between our measurements and the Bigliani types to potentially elucidate any associations with clinical outcomes. However, due to the non-pathological nature of our study design, a direct comparison or application of the Bigliani classification within our current work remains beyond its scope. While our focus was on providing reference values for acromion-related parameters, we recognize the historical significance of the Bigliani classification and its association with shoulder pathologies.

Our reference values are completely independent of individual characteristics such as sex, age, anthropometric data, or dominant hand, making their clinical use straightforward. This raises the hypothesis that behavioral therapy strategies may have limited success in acromial pathologies. There is a certain correlation between the anatomy of the left and right side, at least in terms of the acromion tilt. This could be justified by the assumption that human anatomy is roughly symmetrical. As a potential hypothesis, deviations could be due, for example, to varying degrees of strain during the growth phase. Interestingly, despite focusing on sex differences in subacromial impingement, our findings demonstrate no difference in acromion anatomy, matching the results of Nyffeler et al. or Colegate-Stone et al. [9,16]. In contrast, multiple investigations found a difference between genders in glenoid anatomy [26,27]. We also could not reproduce the findings from Chaimongkhol et al. showing a higher frequency of hooked type acromion in males [13]. Then again, the anatomy of the humeral head seems to show no difference between men and women [28]. This raises the question of why men suffer from this pathology more often than women [29,30], which could be a subject of further investigation, possibly indicating a gender bias or differences in diagnosis. It is crucial to note the regional limitation of our data, which primarily encompasses a German population. Generalizing our findings to diverse populations, including those in Asia or America, should be approached with caution and may require further studies in these regions.

To predict the precise clinical benefit of our reference values, further studies are necessary. The primary aim of this study was to create reference values to aid in deciding whether a patient could benefit from surgical interventions like acromioplasty. We did not find any correlation between pain intensity and single anatomical parameters, which is not surprising given the healthy characteristics of our cohort. Performing similar examinations in a cohort of symptomatic patients and comparing them to our reference values could provide valuable insights. Nonetheless, although subacromial impingement is the most common cause of shoulder pain [31], further potential causes should not be ignored [32,33] but have been minimized by mainly excluding arthritic changes through our methodology. Nevertheless, Chalmers et al. did not find any correlation between acromion morphology in MRI and the likelihood of developing rotator cuff tears, discouraging surgery like lateral acromioplasty [19].

Despite the robustness of our study, certain methodological considerations should be acknowledged. The whole-body MRI captured the acromion in only a few cases, which may be attributed to the risk of a systematic error in excluding participants with a large height or issues with standardization during MRI. That could be reinforced by the fact that the position of the patient in the MRI is different than in X-ray. This is therefore a difference from X-rays of the shoulder joint, which may result in a different scapula alignment and consequently deviations. Although the MRI images have been controlled to ensure correct scapular orientation and minimize potential impacts on measurements, a different scapula orientation could have led to variations. Moreover, the larger layer thickness (5 mm) of the whole-body MRI compared to the standard shoulder MRI thickness (lower than 3 mm) and the poor resolution of the shoulder area are methodological challenges. Additionally, measuring small angles in transversal planes proved difficult with our plug-in, potentially leading to inaccuracies which are shown best in the larger deviation of the interrater variability of acromion slope measurements. Future studies should consider using coronary or sagittal planes for more precise and practical measurements. Nevertheless, the high-quality data from our large collective allowed us to describe the anatomy of the acromion in detail, a feat not previously achieved to this extent, and methodical issues have been approached by interobserver and intraobserver reliability analysis.

This study provides valuable reference values for acromion-related parameters, offering insight into the anatomy of a healthy shoulder. Further investigations are necessary to explore the clinical implications of these reference values and their applicability in patients with acromial pathologies. The findings also raise questions about potential gender biases and the utility of behavioral therapy strategies in managing acromial conditions. Our work contributes to the growing body of knowledge on shoulder anatomy and provides a foundation for future research in this field.

## 5. Conclusions

In conclusion, our comprehensive anatomical analysis of the acromion has yielded valuable reference values for commonly used parameters such as acromion slope, tilt, and the acromion index. These reference values are based on a large and healthy population, providing a solid foundation for clinical use and research in the field of acromial pathologies.

Notably, our findings indicate no significant differences in acromion morphology based on sex, weight, BMI, or dominant hand. This suggests that the reference values are applicable across diverse patient populations, independent of individual characteristics.

While our study sheds light on the normal anatomy of the acromion, further research is necessary to ascertain the clinical implications of these reference values. Prospective investigations are warranted to explore potential associations between acromion morphology and the development of rotator cuff injuries. Additionally, studies comparing symptomatic patients to our reference values could provide valuable insights into the diagnostic and therapeutic utility of these measurements.

The establishment of standardized reference values opens new possibilities for enhancing clinical decision making regarding surgical interventions, such as acromioplasty. By comparing individual patient measurements to these reference values, healthcare providers can make more informed treatment decisions tailored to each patient’s unique anatomy. While MRI proved to be a valid alternative diagnostic tool for assessing acromial issues without any radiation exposure, CT remains the gold standard for examining bony structures and might be used additionally [19].

In summary, our study contributes to the growing body of knowledge on acromial anatomy and provides an essential resource for clinicians and researchers alike. As our understanding of acromial pathologies continues to evolve, these reference values will prove indispensable in advancing patient care and optimizing treatment strategies.

## Figures and Tables

**Figure 2 diagnostics-14-00107-f002:**
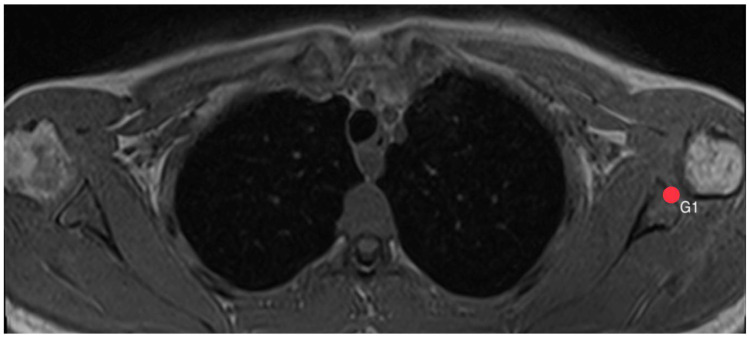
How to measure acromion index: Axial MRI of the thorax depicting the shoulder joint in T1 weighting (3D-GRE VIBE sequence), with a slice thickness of 3 mm. Point G1 indicates the lower end of the glenoid.

**Figure 3 diagnostics-14-00107-f003:**
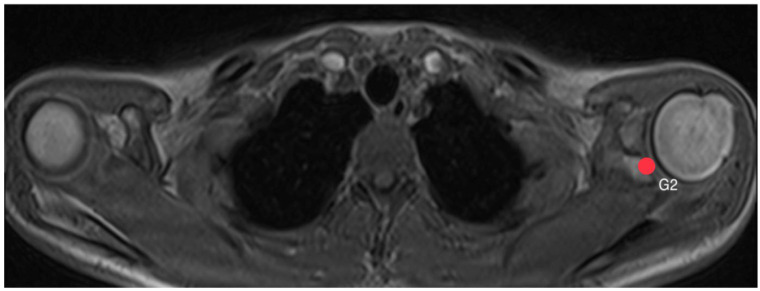
How to measure acromion index: Axial MRI of the thorax depicting the shoulder joint in T1 weighting (3D-GRE VIBE sequence), with a slice thickness of 3 mm. Point G2 indicates the upper end of the glenoid.

**Figure 4 diagnostics-14-00107-f004:**
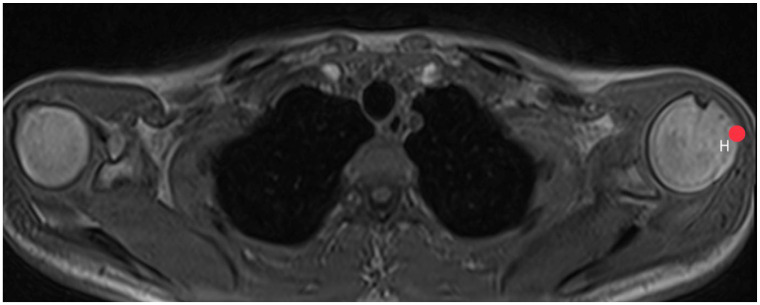
How to measure acromion index: Axial MRI of the thorax depicting the shoulder joint in T1 weighting (3D-GRE VIBE sequence), with a slice thickness of 3 mm. Point H indicates the lateral edge of the humeral head.

**Figure 5 diagnostics-14-00107-f005:**
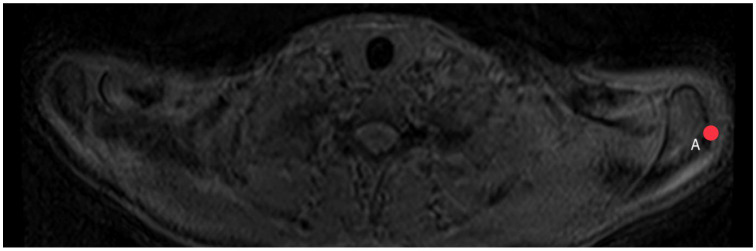
How to measure acromion index: Axial MRI of the thorax depicting the shoulder joint in T1 weighting (3D-GRE VIBE sequence), with a slice thickness of 3 mm. Point A indicates the lateral edge of the acromion.

**Figure 7 diagnostics-14-00107-f007:**
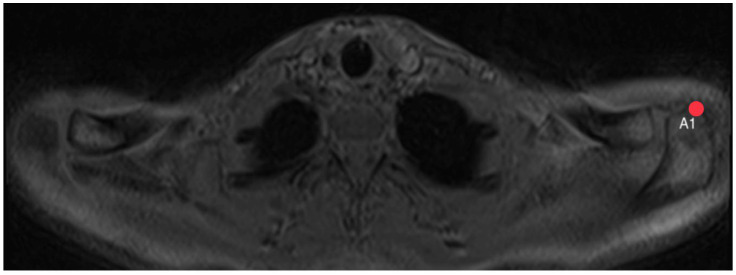
How to measure acromion tilt: Axial MRI of the thorax depicting the shoulder joint in T1 weighting (3D-GRE VIBE sequence), with a slice thickness of 3 mm. Point A1 indicates the ventral end of the acromion’s bottom.

**Figure 8 diagnostics-14-00107-f008:**
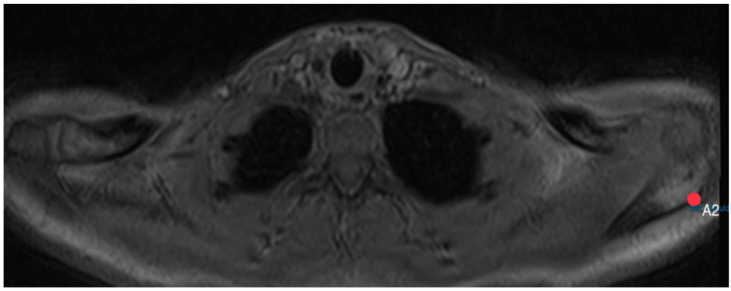
How to measure acromion tilt: Axial MRI of the thorax depicting the shoulder joint in T1 weighting (3D-GRE VIBE sequence), with a slice thickness of 3 mm. Point A2 indicates the dorsal end of the acromion’s bottom.

**Figure 9 diagnostics-14-00107-f009:**
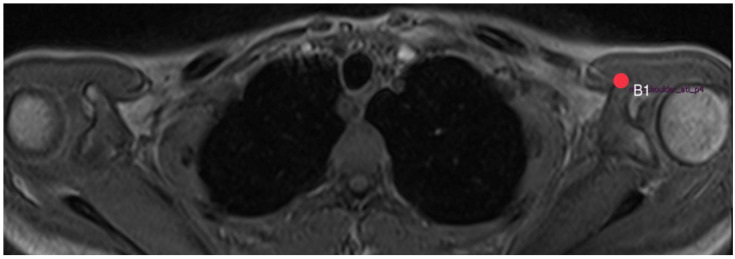
How to measure acromion tilt: Axial MRI of the thorax depicting the shoulder joint in T1 weighting (3D-GRE VIBE sequence), with a slice thickness of 3 mm. Point B1 indicates the lower end of the coracoid process.

**Figure 11 diagnostics-14-00107-f011:**
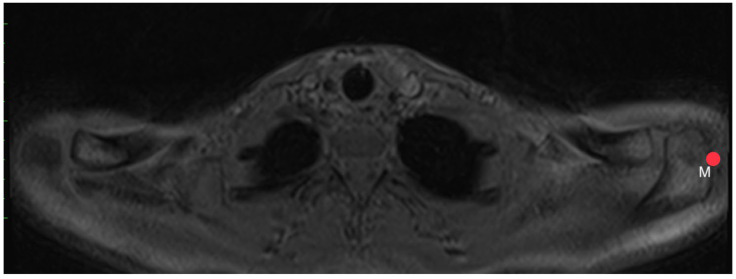
How to measure acromion slope: Axial MRI of the thorax depicting the shoulder joint in T1 weighting (3D-GRE VIBE sequence), with a slice thickness of 3 mm. Point M indicates the middle of the acromion’s bottom.

**Figure 12 diagnostics-14-00107-f012:**
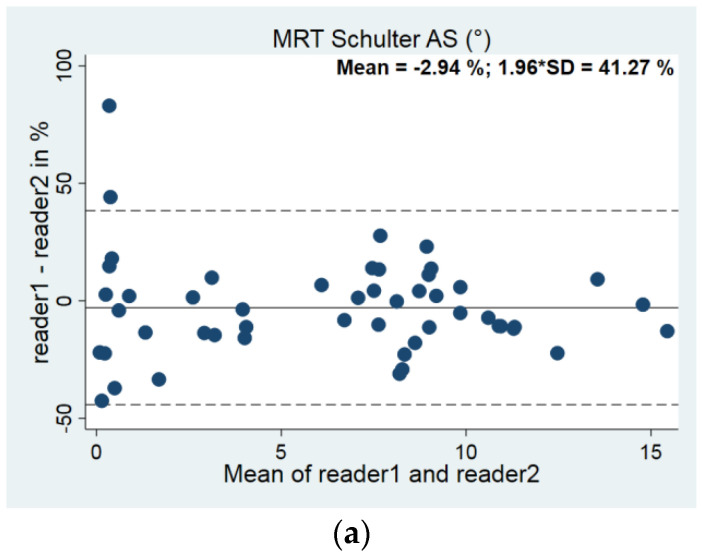
(**a**–**c**). Bland–Altman plots for interrater variability: (**a**) Acromion slope (AS, *n* = 60), (**b**) Acromion tilt (AT, *n* = 85), (**c**) Acromion index (*n* = 60).

**Figure 13 diagnostics-14-00107-f013:**
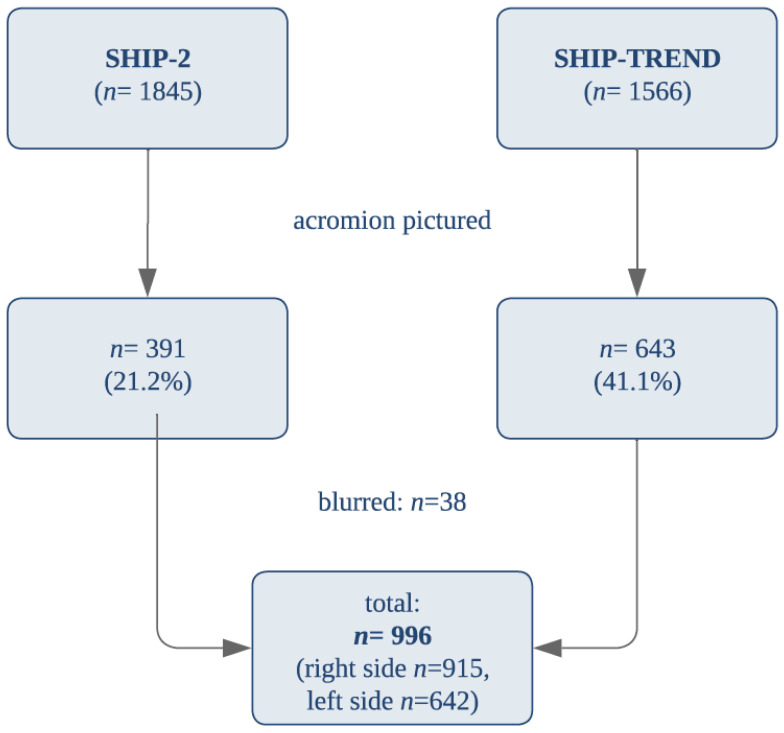
Amount and Composition of Measured MRIs.

**Table 1 diagnostics-14-00107-t001:** Interrater variability values. Number of measurements (*n*), intraclass correlation coefficients (ICC), mean bias (absolute, in % to 360° and their standard deviations (SDs)).

Variable	*n*			ICC	Mean Value		Inter Absolute		Inter %	
	R1	R2	Both		R1	R2	Mean	SD	Mean	SD
Acromion slope	57	55	50	<0.001	6.18	6.43	−0.25	1.04	−2.94	21.05
Acromion tilt	86	85	85	<0.001	31.09	31.45	−0.36	2.43	−1.22	8.09
Acromion index	60	60	60	<0.001	0.494	0.487	0.0066	0.041	2.01	8.15

**Table 2 diagnostics-14-00107-t002:** Correlation between right and left side. Median, 25th and 75th percentile, and Pearson´s correlation coefficient for each parameter.

Parameter	*n* Right Side; Left Side	Right: Median [p25; p75]	Left: Median [p25; p75]	Pearson´s Correlation Coefficient
Acromion slope (°)	876; 606	5.12 [0.99; 8.40]	6.52 [1.24; 9.71]	0.192
Acromion tilt (°)	875; 608	30.96 [27.45; 34.39]	31.19 [28.29; 34.55]	0.529
Acromion index	915; 642	0.53 [0.39; 0.64]	0.48 [0.35; 0.59]	0.339

**Table 3 diagnostics-14-00107-t003:** Differences between genders. Median, 25th and 75th percentile, and *p*-values for the difference between both genders. A *p*-value lower than 0.05 was considered statistically significant (r = right side, l = left side).

Parameter	Female: Median [p25; p75]	Male: Median [p25; p75]	*p*-Value
Acromion slope r (°)	4.66 [0.966; 8.772]	5.47 [1.146; 7.794]	0.575
Acromion slope l (°)	7.10 [1.411; 10.057]	5.88 [1.042; 9.104]	0.202
Acromion tilt r (°)	31.01 [27.46; 34.60]	30.78 [27.42; 34.21]	0.546
Acromion tilt l (°)	31.56 [28.37; 34.64]	30.64 [28.01; 34.39]	0.126
Acromion index r	0.53 [0.425; 0.615]	0.53 [0.437; 0.630]	0.761
Acromion index l	0.50 [0.395; 0.597]	0.52 [0.422; 0.590]	0.412

**Table 4 diagnostics-14-00107-t004:** Correlation between anthropometric parameters and each acromion parameter. A *p*-value lower than 0.05 was considered statistically significant (r = right side, l = left side).

Parameter	Pearson’s Correlation Coefficient	95% Conf. Interval	*p*-Value
**Acromion slope r (°)**			
Height	0.032	[−0.097; 0.162]	0.624
Weight	−0.078	[−0.143; −0.012]	0.021
BMI	−0.264	[−0.465; −0.063]	0.010
**Acromion slope l (°)**			
Height	−0.026	[−0.175; 0.123]	0.729
Weight	−0.020	[−0.101; 0.061]	0.624
BMI	−0.035	[−0.281; 0.210]	0.778
**Acromion tilt r (°)**			
Height	−0.013	[−0.080; 0.055]	0.714
Weight	0.030	[−0.006; 0.066]	0.104
BMI	0.101	[−0.009; 0.210]	0.071
**Acromion tilt l (°)**			
Height	−0.030	[−0.110; 0.050]	0.458
Weight	0.013	[−0.031; 0.056]	0.573
BMI	0.071	[−0.057; 0.199]	0.276
**Acromion index r**			
Height	0.002	[−0.001; 0.004]	0.170
Weight	0.001	[−0.001; 0.002]	0.787
BMI	−0.001	[−0.005; 0.003]	0.663
**Acromion index l**			
Height	0.001	[−0.003; 0.003]	0.889
Weight	0.001	[−0.002; 0.002]	0.936
BMI	−0.001	[−0.005; 0.004]	0.919

**Table 5 diagnostics-14-00107-t005:** Differences between right- and left-handed participants. Coefficients for each parameter considering the declared dominant hand of the participants. A *p*-value lower than 0.05 was considered statistically significant (r = right side, l = left side).

Parameter	Pearson’s Correlation Coefficient	95% Conf. Interval	*p*-Value
Acromion slope r (°)	1.239	[−2.435; 4.914]	0.508
Acromion slope l (°)	−1.025	[−5.510; 3.461]	0.654
Acromion tilt r (°)	0.320	[−1.711; 2.350]	0.757
Acromion tilt l (°)	−0.083	[−2.560; 2.393]	0.947
Acromion index r	0.023	[−0.044; 0.091]	0.501
Acromion index l	0.023	[−0.047; 0.093]	0.516

**Table 6 diagnostics-14-00107-t006:** Differences between participants with pain intensity <3 and >3. Coefficients for each parameter considering the declared pain on a visual analog scale from 1 to 10. A *p*-value lower than 0.05 was considered statistically significant (r = right side, l = left side).

Parameter	Pearson’s Correlation Coefficient	95% Conf. Interval	*p*-Value
Acromion slope r (°)	−1.994	[−4.167; 0.178]	0.072
Acromion slope l (°)	0.287	[−2.379; 2.952]	0.833
Acromion tilt r (°)	−0.696	[−1.871; 0.478]	0.245
Acromion tilt l (°)	−0.240	[−1.723; 1.243]	0.750
Acromion index r	−0.006	[−0.044; 0.032]	0.760
Acromion index l	0.021	[−0.025; 0.068]	0.371

**Table 7 diagnostics-14-00107-t007:** Reference values for each anatomical parameter [with corresponding 95% confidence interval] respecting our excluding criteria (see above) (r = right side, l = left side).

Parameter	*n*	2.5th Percentile	Median	97.5th Percentile
AS r (°)	461	0 [0; 0]	5.48 [3.81; 6.46]	13.15 [12.27; 14.79]
AS l (°)	334	0 [0; 0]	6.48 [5.01; 7.34]	16.27 [14.03; 20.27]
AT r (°)	461	20.26 [18.93; 21.87]	30.83 [30.25; 31.35]	41.70 [39.99; 42.79]
AT l (°)	335	21.97 [20.09; 23.61]	31.62 [30.97; 32.41]	42.01 [40.64; 43.18]
AI r	393	0.26 [0.25; 0.29]	0.52 [0.50; 0.55]	0.72 [0.71; 0.73]
AI l	294	0.25 [0.24; 0.28]	0.50 [0.48; 0.52]	0.72 [0.69; 0.73]

## Data Availability

The data presented in this study are available within the manuscript.

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
