# Peer review of "Establishing Normative Values for Acromion Anatomy: A Comprehensive MRI-Based Study in a Healthy Population of 996 Participants"

_diagnostics, 2024, doi:10.3390/diagnostics14010107_

Round 1

Reviewer 1 Report

Comments and Suggestions for Authors

Material and methods: how were the MRI performed? which was the position of the patient on the table and the position of the arms? It is well known that scapula orientation changes in the erect position since the spine adaption to gravity or in cases of dorsal kyphosis that is easily met in the elderly: how did you obviate for such point? It should be well described in the limitation section.

Could the Authors provide 3 d images showing the measurement performed in order to increase reader's understanding?

Results

Which was the intra and inter-rater reliability value? Could you please include the values?

Line 215: the Authors state a "strong" correlation was found between the left and right side for the acromion tilt. Unfortunately, a correlation coefficient of 0.53 is not considered strong but moderate. Such statement should be changed to "moderate" correlation. Besides, how do the authors explain such results?

Discussion

268-273: the difference with x-rays is that the former are performed in the erect position: please see the considerations above.

The lack of accuracy in measurements performed due to the highlighted limitations is the major drawback of this paper.

Reviewer 2 Report

Comments and Suggestions for Authors

Dear Authors 

Congratulations on tapping the data from the enormous study to establish the normal measurement standards for acromion-based anthropometric studies. However, I have the following concerns that need to be clarified before considering your data for publication. 

1. Despite saying your method is the best you had major limitations in assessing the measurements given the axial sections and large interval MR sequences. Hence, it would be best if you gave a word of caution and not give much stronger wordings in conclusion as the gold standard since CT is always the gold standard for these scenarios. 

2. the interrater reliability and the kappa statistics of the current measures need to be provided and discussed to establish the validity of the current measurements.

3. the regional limitation of your data also needs to be mentioned since it cannot be generalised for the Asian or American population.  

4. since you have not covered on the pathological aspects, you cannot ignore the Bigliani classification to correlate your findings to them 

5. the tables and figures can be combined for clarity and ease of understanding. 

Comments on the Quality of English Language

minor language errors that need rectification 

Author Response

The minor language errors that needed rectification have been adressed as well. 

Round 2

Reviewer 1 Report

Comments and Suggestions for Authors

The Authors correctly addressed all the reviewer's suggestions. 

Reviewer 2 Report

Comments and Suggestions for Authors

Thanks to the authors to clarifying the concerns and addressing them in the revised manuscript 

Comments on the Quality of English Language

Minor language polishing might be done